# Optimization of Deep Architectures for EEG Signal Classification: An AutoML Approach Using Evolutionary Algorithms

**DOI:** 10.3390/s21062096

**Published:** 2021-03-17

**Authors:** Diego Aquino-Brítez, Andrés Ortiz, Julio Ortega, Javier León, Marco Formoso, John Q. Gan, Juan José Escobar

**Affiliations:** 1Department of Computer Architecture and Technology, University of Granada, 18014 Granada, Spain; diegoaquino@correo.ugr.es (D.A.-B.); jortega@ugr.es (J.O.); jleon@ugr.es (J.L.); jjescobar@ugr.es (J.J.E.); 2Department of Communications Engineering, University of Málaga, 29071 Málaga, Spain; marco.a.formoso@ic.uma.es; 3School of Computer Science and Electronic Engineering, University of Essex, Colchester CO4 3SQ, UK; jqgan@essex.ac.uk

**Keywords:** brain-computer interfaces (BCI), evolutionary computing, multi-objective EEG classification, deep learning

## Abstract

Electroencephalography (EEG) signal classification is a challenging task due to the low signal-to-noise ratio and the usual presence of artifacts from different sources. Different classification techniques, which are usually based on a predefined set of features extracted from the EEG band power distribution profile, have been previously proposed. However, the classification of EEG still remains a challenge, depending on the experimental conditions and the responses to be captured. In this context, the use of deep neural networks offers new opportunities to improve the classification performance without the use of a predefined set of features. Nevertheless, Deep Learning architectures include a vast number of hyperparameters on which the performance of the model relies. In this paper, we propose a method for optimizing Deep Learning models, not only the hyperparameters, but also their structure, which is able to propose solutions that consist of different architectures due to different layer combinations. The experimental results corroborate that deep architectures optimized by our method outperform the baseline approaches and result in computationally efficient models. Moreover, we demonstrate that optimized architectures improve the energy efficiency with respect to the baseline models.

## 1. Introduction

Recent years have witnessed the constant growth of computational power, being supported by the development of new hardware and software technologies. Consequently, many problems considered to be unsolvable [1] have been successfully addressed, allowing for the emergence of new lines of research in different fields. In this sense, bioinformatics makes intensive use of tools, such as those in computer science [2], mathematics, and statistics with the aim to analyze, understand, and efficiently use biological signals. DNA and brain activity analysis are typical examples in which the use of Machine Learning methods play an important role in the search for complex patterns [3,4].

Biomedical signal processing usually deals with high-dimensional patterns [5,6,7]. In this context, Brain–Computer Interfaces (BCIs) are systems that identify the patterns from brain activity and send interpretable commands to an electronic device. BCI involves five main steps [8]: (1) signal acquisition of neural activity. (2) Signal preprocessing, which cleans and removes noise from the raw signals. (3) Feature selection (FS) to extract the most significant features from signals. (4) Pattern classification to recognize and categorize brain patterns. (5) The application module provides a feedback to the user from the recognized brain activity pattern.

Brain activity signal acquisition can be classified into three main categories, which are described, as follows: the invasive approach implies surgically implanted technological devices within the human body. This is the most powerful procedure, but it carries many risks due to surgery. Electrocorticography (ECoG) is an example [9], which uses electrodes placed directly on the exposed surface of the brain to record electrical activity from the cortex. Because surgical intervention is needed to place the electrodes, it is considered to be an invasive technique.

On the other side, the non-invasive approach only involves external sensors or electrodes that are placed along the scalp; an example of this procedure is the Electroencephalography (EEG) [10], which registers the electrical brain activity and whose main advantage is the higher temporal resolution [11,12], as compared to Functional Magnetic Resonance Imaging (fMRI).

In this research, the analyzed data correspond to EEG signals because of their practicality and non-invasive character, as well as several advantages over other alternatives [13,14,15,16].

EEG signal classification is a complex, intricate, and challenging task, because EEG samples suffer from low signal-to-noise ratio and the omnipresence of artifacts, which are signals not generated by brain activity. Additionally, the curse of dimensionality problem [17,18] is usually present, due to the nature of biological signals, which produce samples with high-dimensional patterns, and the high cost of the signal registration procedure, which limits the number of EEG samples. The use of feature selection techniques helps to address this issue, which usually results in model overfitting.

Many previous BCI classification techniques make use of FS techniques [19,20,21] to select descriptors from the EEG power bands distribution profile, in order to build the set of selected features, which are used for EEG signal classification. Although this procedure reduces the computational cost that is associated to the feature extraction stage and reduces overfitting in the generated models, it is prone to a loss of information due to a set of unselected (or unknown) features. This way, the accuracy of the classification procedure is highly dependent on the a priori extracted features. However, different filter and wrapper methods have been developed to select the most discriminative feature set. Evolutionary Algorithms (EAs) and multi-objective optimization methods have demonstrated their usefulness in this field [7,20,22,23,24].

EEG signal classification still has room for improvement, and the complexity of the problem makes Machine Learning (ML) techniques appropriate to find the best solutions [25]. ML techniques can help to find the best subset of features, as explained above. However, the most interesting aspect is the possibility of extracting specific, not a priori known, features that maximize the classification performance. In this way, Deep Learning (DL) architectures provide new opportunities to improve the classification performance by enhancing the generalization capabilities of the predictive models that compute specific features during a learning process. In fact, Deep Neural Networks (DNNs) have been used for complex classification problems, outperforming previous classification techniques. Particularly, Convolutional Neural Networks (CNNs) have been successfully applied to this end in many image classification tasks [26,27].

However, an efficient DL architecture requires the correct set of hyperparameters [28]. This is hard to find, because there is not an explicit methodology to do that and the number of hyperparameters is usually high. In this regard, Auto Machine Learning (autoML) refers to a set of methodologies and tools for automatic optimization of Machine Learning models, aiming to generate the best models for a specific problem. A representative example is the process of hyperparameters optimization (HPO), to find the best set of model parameters that provide the best classification performance and minimize the generalization error. Some of the methods that are widely used for HPO are grid search, random search, manual search, Bayesian optimization, Gradient-based optimization, and evolutionary optimization [29,30,31]. More specifically, in the field of DL, [29] implements an iterative process to speed up the HPO, where a DNN is trained and its learning curve is extrapolated using probabilistic methods to predict its performance, which is compared to previous DNN results with different hyperparameter settings. DNN training continues if the predicted performance exceeds records, otherwise it ends immediately. The optimization process continues until the stop criterion is reached. In [30], a proposal using a Genetic Algorithm (GA) for HPO is shown, with an iterative refinement procedure to find the optimal solution in the searching space. As usual in EAs, such as GA implementations, each parameter to be optimized corresponds to a gene in the chromosome used for the genetic representation. In addition, a fitness function is used to evaluate the quality of each solution.

The use of evolutionary computing to optimize or train neural architectures has been previously proposed in different works. Thus, works, such as [32,33], use GAs for the computation of the neural network weights instead of using a gradient descent-based algorithm. Moreover, these works has been assessed in networks with more than 4M parameters. On the other hand, Xie et al. [34] propose the use of GAs to automatically produce neural architectures for image classification, where each solution is codified by a binary vector. Additionally, the EvoCNN method [35] is a method for the optimiztion of CNN networks for image classification. As explained above, the optimization of neural architectures is a current hot research topic, due to (1) the high number of hyperparameters included in a deep network and (2) the absence of clear rules for manual optimization. Hence, this optimization process usually relies on a trial-and-error process that is guided by the designer’s experience.

However, there are different very important aspects to take into account in order to optimize deep architectures. As previously explained, the high number of parameters in deep networks makes them prone to generating overfitted models, with a reduced generalization capability [36]. In particular, models with high complexity and high-dimensional training patterns are more likely to be affected by this problem. Thus, the model performance decreases with new and unknown data. Finding an adequate CNN configuration becomes a challenging process of trial and error. This paper aims to minimize the complexity of the model and maximize the classification accuracy, in order to optimize the generalization capability of the classifier while decreasing the computational burden. More specifically, our proposal implements multi-objective optimization procedures by evolutionary computing for CNNs architectures in order to improve EEG signal classification. This includes the optimization of hyperparameters, such as the number of filters in the convolutional layers, the stride, or the kernel size. Additionally, our implementation is capable of including regularization layers as well as optimizing the regularization parameters. Furthermore, it is worth noting that our implementation produces deep convolutional architectures that are trained from the EEG time series data without using any precomputed feature. This way, the DL architecture is responsible for all stages in EEG classification: feature extraction, feature selection, and classification, allowing for the CNNs to extract the knowledge that is contained in the raw signals to achieve accurate signal classification [37].

The main contributions of the paper are:1.We propose a fully-configurable optimization framework for deep learning architectures. The proposal is not only aimed to optimize hyperparameters, but it can also be setup to modify the architecture, including or removing layers from the initial solutions, covering the inclusion of regularization parameters to reduce the generalization error.2.Architecture optimization is performed in a multi-objective way. That means that different and conflicting objectives are taken into account during the optimization process.3.It is based on multi-objective optimization. Thus, the result is a pool of non-dominated solutions that provide a trade-off among objectives. This allows for selecting the most appropriate solution by moving through the Pareto front.4.The proposed framework uses both CPUs and GPUs to speed up the execution.

In what follows, Section 2 describes the dataset and methods used in this work. This section also includes the description of the optimization framework that was presented in this paper. Section 3 shows the results that were obtained when applying the proposed method to optimize a CNN for EEG classification along with their statistical validation. Moreover, the performance metrics used to evaluate the solutions proposed by our framework are described. Section 4 analyzes the results and improvements provided. At the same time, the power efficiency of the different alternatives is considered. Finally, Section 5 draws the conclusions.

## 2. Materials and Methods

This section includes the description of the dataset used in this research, followed by the definitions of deep neural networks and their main drawbacks, as well as the optimization procedure.

### 2.1. Data Description

The data used in this work were recorded and proposed by the BCI laboratory at the University of Essex, UK [38]. Human participants were selected for balanced gender distribution, a reasonable range of ages, and an appropriate number of participants who were naive to BCI experiments: the 12 chosen subjects of the experiment were 58% female, aged from 24 to 50, and half of them naive to BCI. In addition, the participants were healthy and they were advised to sleep well before data collection. They were paid for their participation and, before the experiment, they gave their informed consent using a form that was approved by the Ethics Committee of the University of Essex.

The EEG-BCI signals are based on Motor Imagery (MI), a paradigm that uses a series of brief amplifications and attenuations conditioned by limb movement imagination, called Event-Related Desynchronization (ERD) and Event-Related Synchronization (ERS).

Each EEG pattern obtained is a time series consisting of 5120 samples, which were recorded at the sampling rate of 256 Hz. Thirty-two electrodes were placed on the scalp during data collection, but the 15 electrodes that are shown in Figure 1 were selected for EEG feature extraction and classification, which was determined based on BCI performance optimization.

Each dataset is composed of 178 training patterns and 179 testing patterns, and each pattern is labelled according to the corresponding BCI class (imagined left hand movement, imagined right hand movement or imagined feet movement) which are detailed in Table 1. Three subjects, coded as 104, 107, and 110, were selected, as they provided the best performance in previous works dealing with EEG-BCI signal classification [7,20]. Thus, our aim was to use the optimization framework to improve the best results that were previously obtained by other methods.

### 2.2. Deep Neural Networks

Deep Learning architectures are essentially neural networks, but different layers and learning algorithms have been developed to solve specific problems [39]. For instance, convolutional layers have demonstrated their ability to extract relevant features for image and time series classification. Moreover, features that are learnt from deep architectures are retrieved at different abstraction levels. However, the most important is that these features are computed by a learning process that modifies the network parameters to minimize the output of a loss function.

The so-called (artificial) neuron is the basic processing unit of a neural network, which computes a simple mathematical function *z*. The output of this function represents the activation output of the neuron. In the case of linear activation, *z* can be expressed, as defined in Equation (Equation 1), where $w_{i}$ is the weight that is associated to the *i*-th input of the neuron, $a_{i}$ is the *i*-th input, and *b* is a bias term.
(1)$$z=b+\sum_{i=1}^{n}a_{i}\;\cdot \;w_{i}$$

However, different activation functions are used, as they allow for the computation of the gradients during backpropagation, along with the creation of deep architectures. Moreover, vanishing and exploding gradients are well known problems that are related to the use of unbound activation functions [40], making the learning algorithm unstable or not converging. These effects can be mitigated using bounded activation functions that limit the gradient values. Thus, a range of activation functions is usually used and combined at different layers. The ones considered in this work are the following (see Figure 2):*Sigmoid:* is a logistic function, where the input values are transformed into output values within a range of (0,1). The function definition is given by Equation (Equation 2).
(2)$$f\left(z\right)=\frac{1}{1+e^{-z}}$$*Scaled Exponential Linear Unit (SELU):* ensures a slope larger than one for positive inputs. If the input value *z* is positive then the output value is multiplied by a coefficient α. Otherwise, when the input value *z* is equal or less than 0, then the coefficient α multiplies the exponential of the input value *z* minus the α coefficient, and, finally, the result is multiplied by a coefficient λ. The function definition is given by Equation (Equation 3).
(3)$$f\left(z\right)=\lambda \left\{\begin{array}{cc} z & \mathrm{if}\;z>0\\ \alpha e^{z}-\alpha & \mathrm{if}\;z\leq 0 \end{array}\right.$$*Hyperbolic tangent (TanH):* is a useful activation function with a boundary range of (−1,1), which allows for efficient training. However, its main drawback occurs in the backpropagation process, due to the vanishing gradient problem that limits the adjustment of the weight value. Equation (Equation 4) provides the function definition.
(4)$$f\left(z\right)=\frac{2}{1+e^{-2z}}-1$$*Rectifier Linear Unit (ReLU):* a commonly used function, where, if the input value *z* is equal to or less than 0, then *z* is converted to 0. In the case of a positive input *z*, the value is not changed. The function definition is given by Equation (Equation 5).
(5)f(z)=max(0,z)*Leaky ReLU (LReLU):* similar to ReLU. The difference occurs when the input value *z* is equal to or less than 0, then *z* is multiplied by a coefficient α which is usually within the range (0.1,0.3). Equation (Equation 6) provides the function definition.
(6)$$f\left(z\right)=\left\{\begin{array}{cc} z & \mathrm{if}\;z>0\\ \alpha z & \mathrm{if}\;z\leq 0 \end{array}\right.$$*Exponential Linear Unit (ELU):* compared to its predecessors, such as ReLU and LReLU, this function decreases the vanishing gradient effect using the exponential operation $e^{z}$. If the input value *z* is negative, then $\left(e^{z}-1\right)$ is multiplied by a coefficient α in the common range of (0.1,0.3). The function definition is given by Equation (Equation 7).
(7)$$f\left(z\right)=\left\{\begin{array}{cc} z & \mathrm{if}\;z>0\\ \alpha (e^{z}-1) & \mathrm{if}\;z\leq 0 \end{array}\right.$$
where the horizontal and vertical axes in Figure 2 represent the inputs and outputs of the activation functions, respectively.

DNN learning is implemented by using the backpropagation algorithm, a supervised learning method that is usually considered for classification and regression problems, which uses the gradient of the loss function with respect to the weights of the network, enabling the weights adjustment produce expected output values with given input signal. The process begins from the output layer, continues recursively through the hidden layers, and concludes when the inputs layer is reached. Each neuron of the network is responsible for a percentage of the total error. Thus, the neuron error is computed and propagated to the entire network by using the chain rule for derivatives.

#### Convolutional Neural Networks

CNNs are deep neural networks that efficiently process structured data arrays, such as spatial and temporal ones [25]. In a nutshell, CNNs architectures are multi-layer architectures allowing for the hierarchical learning of features. The layer considered as essential foundation of CNNs is the so-called convolutional layer [40]. The convolution operation (denoted as (∗)) between two functions f(x) and g(x) produces a third function s(x). The function f(x) corresponds to the input and g(x) to the filter, while s(x) corresponds to the feature maps that were obtained as a product of convolving f(x) and g(x), as defined in Equation (Equation 8).
(8)$$s\left(x\right)=\left(f*g\right)\left[x\right]=\sum_{i=1}^{n}f\left(i\right)\;\cdot \;g\left[x-i\right]$$
where *x* is a discrete variable, i.e., arrays of numbers, and *n* corresponds to the filter size.

Moreover, there is another layer specially used in CNNs, called the pooling layer, used to reduce the dimensionality of the data in order to decrease the computing requirements. Pooling can be performed by averaging the samples in a specific window (average pooling) or taking the maximum value (max pooling). On the other hand, variants of the standard convolutional layer have been devised in recent years, especially for those problems with multiple input channels. This is the case of the depthwise convolution, which allows for applying each filter channel only to one input channel. Additionally, since the depth and spatial dimension of a filter can be separated, it helps to reduce the number of parameters by means of Depthwise separable convolutions. It is worth noting that one-dimensional (1D) convolutions can be carried out by conv1D or conv2D Keras functions. In Conv1D, the convolution operates in the only dimension defined, whereas, in conv2D, the convolution operates on the two axes defining the data. Thus, conv1D of a signal of size *a* is equivalent to conv2D of a signal of size a×1. Because the original EEGNet network uses conv2D, we kept conv2D in the architecture to be optimized.

Convolutional layers [40] configured in the optimization process described here are listed below:**Convolution 2D**: exploits spatial correlations in the data. This layer can be composed of one or more filters, where each one is sliding across a 2D input array and performing a dot product between the filter and the input array for each position.**Depthwise Convolution 2D**: aims to learn spatial patterns from each temporal filter allowing for feature extraction from specific frequencies of the spatial filters. This layer performs an independent spatial convolution on each input channel and, thus, produces a set of output tensors (2D) that are finally stacked together.**Separable Convolution 2D**: aims to reduce the number of parameters to fit. This layer basically decomposes the convolution into two independent operations: the first one performs a depthwise convolution across each input channel, while the second one performs a pointwise convolution that projects the output channels from the depthwise convolution into a new channel space.

At the same time, we include the possibility of using different pooling alternatives as well as batch normalization, as they may help to speed up and stabilize the learning process:**Average Pooling 2D**: aims to reduce the representation of the spatial size of each channel. This layer takes the input data array and calculates the average value from all values of each input channel. This way, it generates a smaller tensor than the corresponding input data.**Batch Normalization [41]**: aims to achieve fixed distributions of the input data and address the internal covariate shift problem. This layer performs the calculation of the mean and variance of the input data.

### 2.3. Overfitting

Machine Learning is a process of feature extraction and pattern identification from a given set of data. It allows for the generation of a model with high capacity of generalization and, therefore, provides autonomy to the computer in order to perform specific tasks. However, the capabilities of generalization can be affected when the fit of the model is extremely close to the data, to the point of mistaking existing noise for relevant information. This problem is known as overfitting [42].

Overfitting can occur in several circumstances [18,43], such as having few data samples for the model training process, using a neural network with a large numbers of parameters, overtraining of the model, having data with considerably more features than available samples [17], etc.

DL models are likely to suffer from overfitting by nature, since they are usually composed of a large number of parameters. This way, optimizing the models to obtain shallower architectures plays an important role to avoid overfitting. However, the number of layers (and, as a consequence, the number of parameters) of a model greatly depends on the hyperparameters that have to be adjusted for a specific task [44]. Beyond the selection of the model with the minimum number of parameters, overfitting can still be present due to high model variance that is produced by the absence of enough data samples. In this case, regularization methods can be used to penalize network weights during training, while the backpropagation algorithm forces the network to generate the correct output. Among the regularization methods, Dropout [45] is an explicit technique that is used for that purpose, which makes learning difficult by deactivating some network neurons at random during the training process, with the aim of preventing over-adaptation by neuronal units, thereby ensuring generation of more robust models. Kernels and activations can be also regularized by the introduction of $\ell_{1}$ or $\ell_{2}$ regularization term that penalizes large weights or large activations. Examples of $\ell_{1}$ and $\ell_{2}$ regularized loss expressions are shown in Equations (Equation 9) and (Equation 10), respectively.
(9)$$\ell_{1}=error\left(y-\hat{y}\right)+\lambda \sum_{i=1}^{N}\left|w_{i}\right|$$
(10)$$\ell_{2}=error\left(y-\hat{y}\right)+\lambda \sum_{i=1}^{N}{∥w_{i}∥}^{2}$$
where *y* is the data to be predicted, $\hat{y}$ is the prediction made by the model, λ is a regularization parameter, and $w_{i}$ is the i-th weight component.

On the other hand, early stopping [46] of the training algorithm is an implicit regularization method that helps to prevent the problem that is addressed in this section. Early stopping is triggered by a low loss value over a number of training epochs in the validation data set.

### 2.4. Multi-Objective Optimization

Multi-objective optimization problems [47] are usually present today’s world. These problems can be solved using analytical methods or classical numerical methods. Moreover, several types of heuristic search algorithms have been proposed to address these problems. There are four categories of optimization algorithms, which are inspired by Biology, Physics, Geography, and Social culture.

Biologically inspired algorithms try to mimic evolutionary processes or behaviors found in nature. EAs are inspired by the improvement of individuals in the population through successive generations by means of a natural selection process: the best individuals are selected and reproduced with each other, producing an offspring for the next generation, while the worse ones are withdrawn from the population.

The optimization approach that is presented in this work is based on a multi-objective optimization procedure, which aims to find the vector $x=[x_{1},x_{2},...,x_{n}]$ that optimizes a function vector f(x), whose components $\left(f_{1}\left(x\right),f_{2}\left(x\right),...,f_{m}\left(x\right)\right)$ represent the objectives to optimize. Multi-objective optimization often has objectives in conflict, which results in a set of non-dominated solutions, called Pareto optimal solutions. In a given Pareto front, when all of the objectives are considered, every possible solution is equally efficient. Therefore, it is possible to choose the most convenient solution for a given scenario.

#### 2.4.1. NSGA-II Algorithm

Non-dominated Sorting Genetic Algorithm-II (NSGA-II) [48,49] has been implemented to deal with the multi-objective optimization problem that is considered in our hyperparameter searching problem. Genetic operators and the codification of the individuals are specific for the problem at hand. This work involves a supervised classification problem, where the individual and its chromosomes correspond to the architecture of a convolutional neural network, which is in charge of the signal classification. The performance evaluation of each individual is determined by the complexity of the CNN and its classification accuracy. Algorithm 1 shows the pseudo-code of the NSGA-II.
**Algorithm 1:** Pseudo-code of the Multi-objective Optimization Procedure for Deep Convolutional Architectures using NSGA-II [48,49].
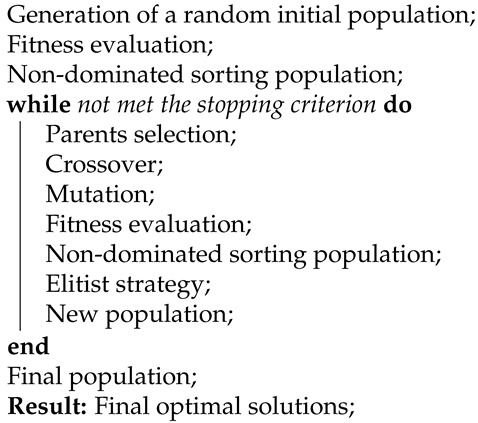


#### 2.4.2. Convolutional Neural Networks for EEG Data Classification

Although CNNs are the most popular neural networks for image and video classification, they can also be used with time series data. In this case, 1D convolutions are used instead of 2D convolutions. An input channel here corresponds to each EEG electrode, since it corresponds to a different signal source. However, EEG signals are different from images by nature, and so are the features to be extracted. Moreover, as a result of the high number of channels in comparison to image data (usually, only three channels in RGB images), it is necessary to use more specific convolutional layers that: (1) make it possible to perform convolutions in the channel axis and (2) help to reduce the number of parameters (i.e., while using separable convolutions).

### 2.5. EEGNet: CNN for EEG Classification

EEGNet [26] is a CNN for EEG-BCI signal classification, which encapsulates the feature extraction step using Depthwise and Separable convolutions. The architecture of the EEGNet consists of three blocks: two of them are dedicated to the implicit feature extraction step and the last one implements the classifier. Figure 3 shows the EEGNet architecture.

Block 1 is dedicated to feature extraction and involves two convolutional layers. The first layer (Convolution 2D) captures information from the temporal frequencies of the input data, while the second layer (Depthwise Convolution) is dedicated to extract information from spatial frequencies. After each convolutional layer, the output values are normalized. The next layer in the block implements the activation function to introduce non-linearity into the output values. These values are then passed through the average pooling layer, reducing the size of the input to the next layer. Finally, a dropout layer is included for regularization.

Block 2 consists of a separable convolutional layer to reduce the number of parameters with respect to a standard convolutional layer. The output values are then normalized and activated by a non-linear function. An average pooling layer again reduces the output size. As in Block 1, dropout is used for regularization. Finally, the input data are collapsed into a one-dimensional array to prepare the input to the classifier.

Block 3 (Classification Block) implements the classifier with a fully-connected layer and uses a softmax activation function to provide the probability of the activation of each neuron at the output.

### 2.6. Performance Evaluation

In this section, we present the performance metrics that were used in this work to evaluate the solutions provided by the optimization procedure. The fitness function of the optimization process includes two objectives: Kappa Index and the number of CNN parameters, which, together, determine the quality of each individual solution. Definitions of these metrics are detailed below:**Pareto front [50]:** multi-objective optimization problems consider several objectives at the same time. Therefore, it is usually not possible to obtain an optimal solution that satisfies all the conditions. Instead, the optimization process provides a set of non-dominated solutions with different trade-offs among the objectives: the Pareto optimal solutions.**Kappa Index [51]:** is a statistic measure that is used for multi-class classification problems or imbalanced class problems. The Kappa Index *k* is defined as:
(11)$$k=\frac{p_{0}-p_{c}}{1-p_{c}}$$
where $p_{0}$ is the observed agreement and $p_{c}$ is the expected agreement. The Kappa Index value is always less than or equal to 1.**CNN Parameters [25]:** They are weights changing during the training process that are also known as trainable parameters.

### 2.7. Proposed Optimization Framework and Application to EEG Signal Classification

The main goal of the optimization framework that is presented in this paper is to find neural network architectures with low complexity and high classification accuracy. To this end, the proposed optimization framework implements the following main components. Figure 4 shows a block diagram with the components of the framework and how they interact during the optimization process.

The key component is NSGA-II, which is the considered EA to perform the multi-objective optimization process. This choice was motivated by the fact that NSGA-II can achieve a good performance with two objectives [52], which makes it a suitable option for the problem at hand.

The framework also includes a database to register the different entities, including the main building blocks of neural networks, such as layers, parameters of each layer, as well as the architecture of the neural model to be optimized and the results that were obtained during the optimization process. Moreover, the ranges of the different parameters to be optimized (i.e., the values that each gene can take during the optimization process) are also configured in the database and are further used as restrictions in the EA.

The procedure that is explained here is applied to the optimization of the EEGNet, which has been used as a baseline. This way, the network entities (layers, regularizers, normalization layers, etc.) and hyperparameters that are included in the EEGNet have been registered in the configuration database.

Figure 5 shows the chromosome generation process using the restrictions stored in the database, specifically for the case of convolutional layers. This way, a *flatten* layer is always included to prepare the output of the convolutional layers to feed the classifier, which, in turn, is a perceptron-like (fully-connected) network. As explained above, the optimization process is implemented by the NSGA-II multi-objective evolutionary optimization algorithm. The chromosome that is shown in Figure 5 depicts the codification of the solutions, where the variables are optimized. These include parameters of the convolutional layers: kernel size, stride, number of filters, and regularization parameters. Moreover, the use of dropout is also activated by a specific gene, as well as the activation function that is used to implement the non-linearity.

Figure 6 shows a breakdown of a chromosome into its constituent genes. This codification allows for the selection of the different parameters indicated in the figure, according to the layers that were used in the baseline network. This chromosome is configured in the database, depending on the restrictions imposed during the optimization process.

The solutions along partial results of the optimization process can be tracked, since they are stored in a structured, standard *PostgreSQL* [53] database.

Genes composing the chromosome are of integer type, as shown in Figure 6. This codification allows for speeding up the searching process and limiting the range of each gene, by means of a look-up table, where each entry is codified as an integer value.

#### Optimization Process and CPU-GPU Workload Distribution

The main goal of the optimization process is to increase the classification performance while decreasing the number of parameters. These are opposed objectives, since a larger network will usually achieve a better fit to the training data. However, our goal is to improve the generalization performance, and larger networks will also be more prone to overfitting. This way, as it has been said before, the fitness functions that are used to evaluate the solutions are the Kappa Index and the total number of parameters of the neural network. Moreover, the proposed framework has been implemented as a parallel application by distributing the different tasks among the processors that are available in the computing platform: a node of a cluster that comprises CPU cores and Graphical Processing Units (GPUs). Specifically, the EA has been executed on a Intel Xeon(R) E5-2640 v4 CPU, while the evaluation of the solutions takes place in the TITAN Xp GPUs. It is noteworthy that the fitness evaluation of a solution requires training and validating a neural network, which are time consuming tasks that can be accelerated by taking advantage of the parallelism implemented by GPUs and the several cores included in a multi-core CPU. Additionally, it is possible to configure the specific combination of CPUs and GPUs to be used in the configuration database of the framework. This way, we can take advantage of the available computational resources, scaling the processing time.

The initial population consisting of a set of random solutions is evolved by applying the genetic operators, as explained in Section 2.4.1. The evaluation of each solution provides the non-dominated solutions. Thus, the best (non-dominated) solutions in terms of the objectives to be optimized are selected for applying the crossover and mutation operators. Table 2 lists the parameters used for the NSGA-II algorithm. The developed optimization framework stores the set of solutions that survived after each generation in the database, along with the respective values of the fitness functions.

We implemented a look-up table where each entry is codified as an integer value in order to speed up the searching process and limit the range of each gene, as indicated in Figure 6. This way, we used the popular Simulated Binary Crossover (SBX) [48], which tends to generate offspring near the parents. This crossover method guarantees that the extent of the children is proportional to the extent of the parents [48].

Regarding the execution time, it needs 116.99 h for subject 104, 139.02 h for subject 107, and 142.81 h for subject 110, while using the computing resources detailed above.

## 3. Results

In this section, we present the results that were obtained for EEGNet optimization, which is used as a baseline in this work. This way, we describe the solutions provided by the methodology implemented in the framework described above and the performance of different optimal solutions according to the Pareto front (the performance of non-dominated solutions, depending on the objectives). Finally, the results are statistically validated.

### 3.1. Data Split for Performance Evaluation

Section 2.1 describes the EEG dataset used in this work. As explained in that section, it consists of two data files per subject: one for training and one for testing. The training process carried out during the optimization is performed by using the training data file, which in turn is split into two sets containing random samples but keeping the label distribution. The first set is composed of 90% of the training samples, while the remaining 10% is used to validate the solutions by means of the fitness functions. Moreover, 15 bootstrap resampling iterations were carried out to estimate the standard error of the models, by extracting from the training data file different subsets for training and validation in each iteration. The test data were always the same and only used after the EEGNet was optimized using the training and validation data (provided in the test data file to this end). All of the experiments were carried out using the Adam optimizer [54] for network training, which is an adaptive learning rate optimization algorithm that was specifically developed for training deep neural networks.

### 3.2. Experimental Results

Optimization experiments taking the EEGNet network as a starting point were carried out. The procedure that was implemented in our framework was independently applied to EEG training data corresponding to three human subjects, coded as 104, 107, and 110. As a result of these three executions of the optimization method, we obtained the corresponding Pareto fronts shown in Figure 7. The Pareto front is the geometric place of the non-dominated solutions, selected from the set of points corresponding to the solutions generated during the evolutionary process, as explained in Section 2.6. In Figure 7, the point that corresponds to the solution providing the trade-off between both objectives (Kappa Index and the number of parameters of the model) is highlighted with a red dot. Additionally, Table 3 details the models corresponding to these solutions, where all of the layers composing each model are indicated. The optimization procedure does not only select the hyperparameter values, but can also decide whether regularization is used or not as well as the regularization method, as explained in Section 2.7. In Table 3, *None* means that this solution is not using that component or layer.

The trade-off points of the Pareto fronts that are shown in Figure 7 correspond to solutions with the hyperparameters detailed in Table 4, where the solutions with optimized hyperparameters provide a higher classification performance than the non-optimized ones. At the same time, the trade-off point of each Pareto front corresponds to the architectures shown in Table 3.

Moreover, Figure 7 graphically depicts the models corresponding to the trade-off solutions.

The models that were produced by the optimization framework were trained using the whole training data file (without the validation split) and then tested using the testing data file. The results that are provided in Table 4 show a clear improvement in the accuracy obtained from the optimized models with respect to the original (baseline) EEGNet model [26] and the DeepConvNet [26]. These results show that the optimized models achieve an average improvement in the Kappa Index of 43.9%, 87.2%, and 27.5% with respect to the original EEGNet, for the subjects 104, 107, and 110, respectively.

Figure 8 depicts the comparison among different EEGNet models for the data of the three subjects that were used in this work. The optimized version always outperformed the baseline networks, as can be seen. The EEGNet-based networks compared in Table 4 mainly differ in the number of filters used in the lower layers: while the so-called DeepConvNet [26] implements a deeper architecture with five convolutional layers and a higher numbers of filters. This is the main reason for the lower number of parameters in the EEGNet baseline.

#### Power Efficiency of the Optimized Solutions

In this section, we analyze the power consumption of the different networks that were used in this work. This aims to evaluate the power efficiency of the optimized network with respect to the ones that were taken as baseline. Figure 9 shows the power profile when the networks are trained using EEG data from subjects 104, 107, and 110. The optimized alternative considerably reduces the instantaneous power consumption and, thus, the average power, as shown in this Figure. Table 5 shows the average power consumption of the different evaluated models, where the optimized model requires less power consumption during the training.

The results obtained regarding the power consumption are as expected due to the reduction in the number of parameters of the optimized model.

## 4. Discussion

Deep Learning architectures have demonstrated their competitive classification performance in many applications. Specifically, some types of networks, such as CNN, have outperformed classical statistical learning classifiers in some applications, such as image classification. This is the case of convolutional neural networks, which are widely and successfully used in image classification tasks. However, the explicit characteristics of each model depend on the problem being tackled. This has led to the development of a wide range of CNN networks, which, although containing convolutional layers, differ in terms of their combination and the hyperparameters used. In this way, autoML techniques contribute to the Deep Learning field with procedures to automatically develop or optimize existing networks to solve a specific problem. This is the case of EEG signal classification, where usual CNN-based architectures for image processing are not appropriate due to the especial characteristics of the EEG signals. Furthermore, the inclusion of layers that allow for spatial and temporal convolutions adds new hyperparameters that are difficult to select. The optimization framework that is presented in this paper contributes with a flexible alternative to optimize existing networks to improve the classification performance on a specific problem. The proposal is based on multi-objective evolutionary optimization, which tries to select the best architecture according to predefined and configurable objectives. In this paper, we use two performance metrics: the Kappa Index to measure the multiclass classification performance, and the number of parameters, which forces the algorithm to select solutions with a smaller number of parameters. The optimization of these metrics is based on the validation results, aiming to enhance the generalization capabilities of the optimum model. The proposed framework has been assessed using EEG data for BCI, composed of 15 channels. Subsequently, a previously developed DL network for EEG-BCI classification, called EEGNet, has been used as a baseline to optimize its performance. The results that are shown in Figure 8 show clear improvements with respect to the original network, achieving up to a 87% improvement with up to 33% fewer trainable parameters. Notably, the optimization method included regularization layers that act on kernels and activation functions, whose need is motivated by the improvement on the generalization capabilities of the network (i.e., to reduce overfitting). Additionally, the results have been statistically validated while using the classification outcomes that were obtained during the bootstrap iterations by means of a Kruskal–Wallis hypothesis test, giving small enough *p*-values (<10${}^{-7}$) to demonstrate the superiority of the optimized solution.

On the other hand, we conducted experiments to measure the power consumption of the different networks in this paper during the training process with data from subjects 104, 107, and 110 from the BCI dataset of the University of Essex. The obtained results demonstrate that the reduction in the number of parameters of the network directly impacts its energy efficiency. This effect is due to the lower GPU usage during training and the shorter training time that is required for the network to converge. As a consequence, the objective aimed to minimize the number of parameters is also minimizing the power consumption of the final architecture.

## 5. Conclusions and Future Work

In this paper, we present a multi-objective optimization framework for optimizing Deep Learning architectures that are based on evolutionary computation. The proposed method is not only intended for hyperparameter tuning, but also for enabling or disabling some layers, such as those that implement regularization. As a result, the architecture of the generated models can be different from the original one. On the other hand, the possibility of using separable convolutions allows for the algorithm to select solutions with similar performance, but with fewer parameters. Consequently, these models are less prone to overfitting the data, exhibiting a lower generalization error. The proposed optimization framework has been assessed using EEG data and the deep neural network EEGNet as a baseline. The effectiveness of the proposal is demonstrated in the experiments performed, showing improvements of up to 87% in the classification performance (Kappa Index) with respect to the EEGNet base model. At the same time, the optimized models are composed of fewer parameters, resulting in shallower networks, which, in turn, are less prone to overfitting. The presented framework can be configured to modify the search space, allowing for the use of more layer types. This provides an appropriate arena to use it with a wide range of problems, not only those that are based on CNN networks. As future work, we plan to improve the framework by allowing the construction of a neural network by itself according to a set of restrictions configured in the database. At the same time, the energy that is consumed by the evaluation of each solution during the training stage will be added as an optimization objective, trying to generate networks that use not only a smaller number of parameters, but also layers that require less computational resources. The reduction in the number of parameters directly impacts on the energy efficiency of the optimized solutions, as demonstrated in Section 3.2.1 and further discussed in Section 4. The main limitation to this end is related to the current parallelism level. Although the proposed framework can be setup to explore a wide search space by relaxing some of the restrictions, it is limited, in practice, by the execution time (as usual in evolutionary computation). In fact, the framework can be used to modify not only the hyperparameters, but also the architecture, including or removing layers in the quest of the best solution. As we consider this part a very interesting research direction, the optimization of the CPU-GPU parallelism is part of our future work, aiming to use the tool for the automatic generation of deep architectures.

## Figures and Tables

**Figure 1 sensors-21-02096-f001:**
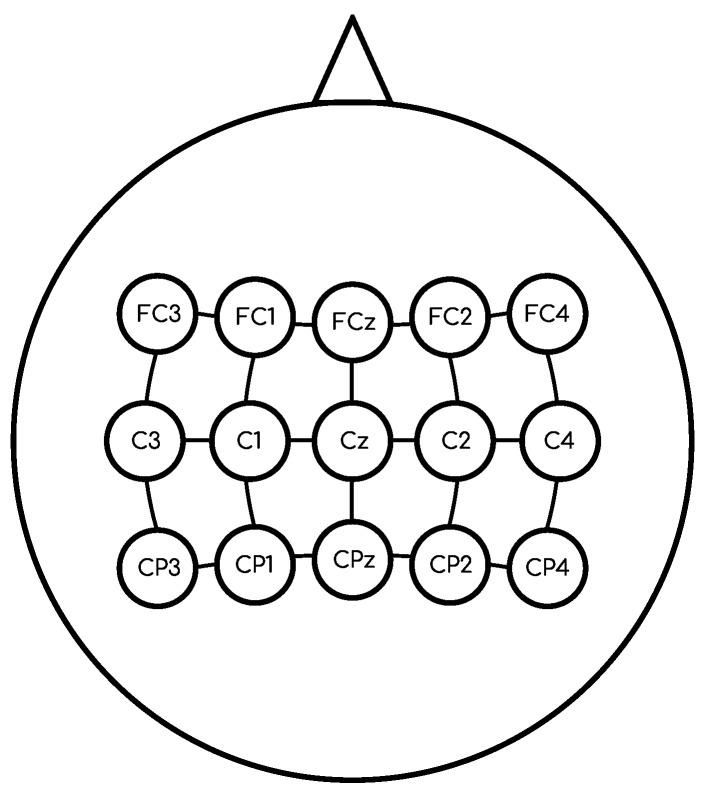
Electrode positions according to the 10–20 coordinate system used for the acquisition of the Electroencephalography (EEG) signals.

**Figure 2 sensors-21-02096-f002:**
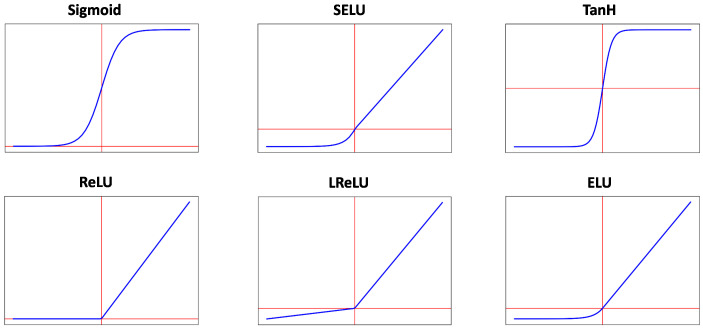
Sigmoid, Scaled Exponential Linear Unit (SELU), Hyperbolic tangent (TanH), Rectifier Linear Unit (ReLU), Leaky ReLU (LReLU), and Exponential Linear Unit (ELU) activation functions.

**Figure 3 sensors-21-02096-f003:**
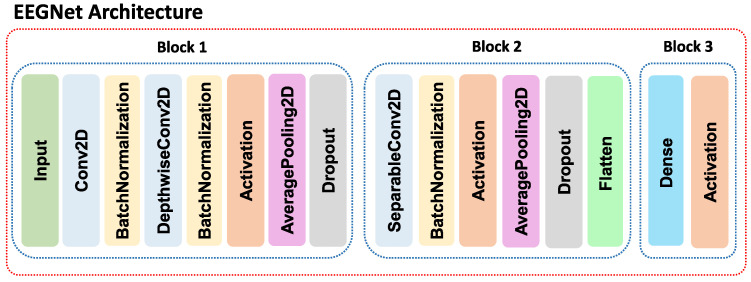
EEGNet baseline architecture.

**Figure 4 sensors-21-02096-f004:**
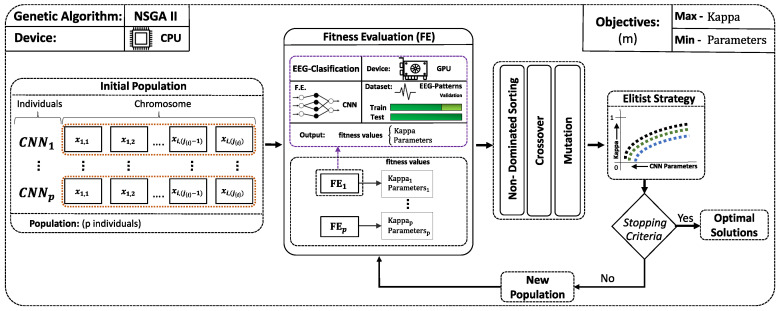
Scheme of the implementation for the evolutionary multi-objective procedure for CNN optimization. $x_{{ij}_{\left\{i\right\}}}$ indicated the *j*-th gene used to code the parameters of the *i*-th layer.

**Figure 5 sensors-21-02096-f005:**
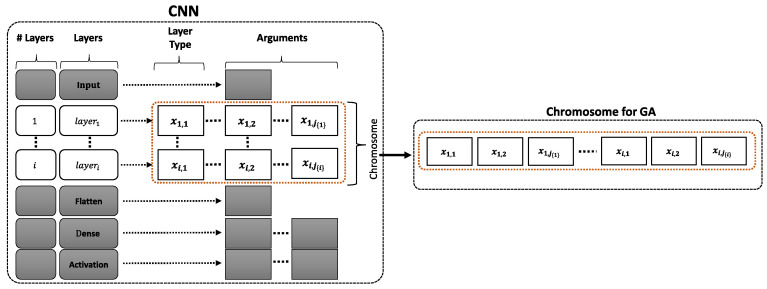
Chromosome generation process using the restrictions that were previously configured in the framework database. $x_{{ij}_{\left\{i\right\}}}$ indicated the *j*-th gene used to code a parameter of the *i*-th layer. Gray shaded boxes indicate the fixed layers not included in the optimization process.

**Figure 6 sensors-21-02096-f006:**
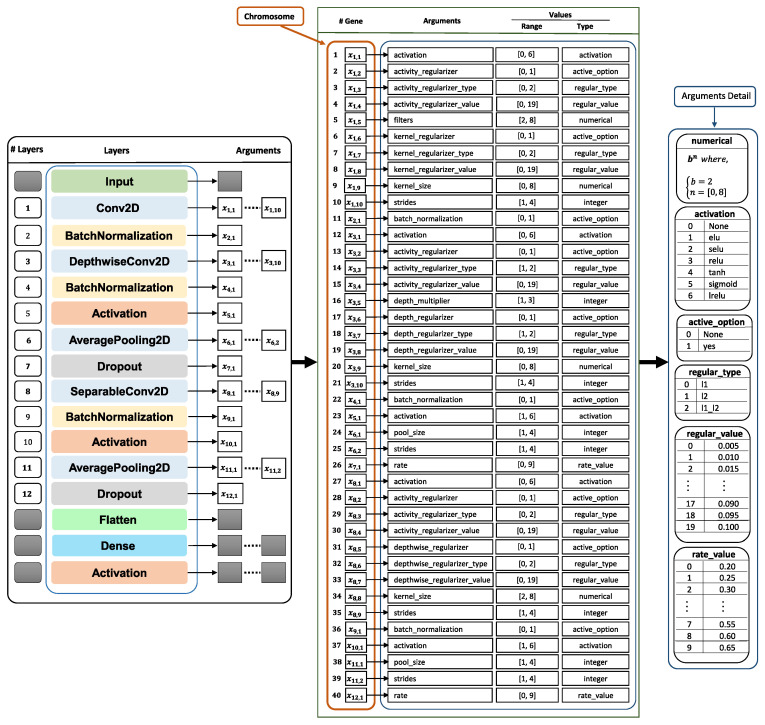
Detailed view of the chromosome that codifies the solutions for evolutionary optimization algorithm (NSGA-II). Figure shows the values used in the optimization of EEGNet architecture shown in this work. However, specific ranges, activation functions, regularization types, etc. can be configured in the database, depending on the specific optimization problem.

**Figure 7 sensors-21-02096-f007:**
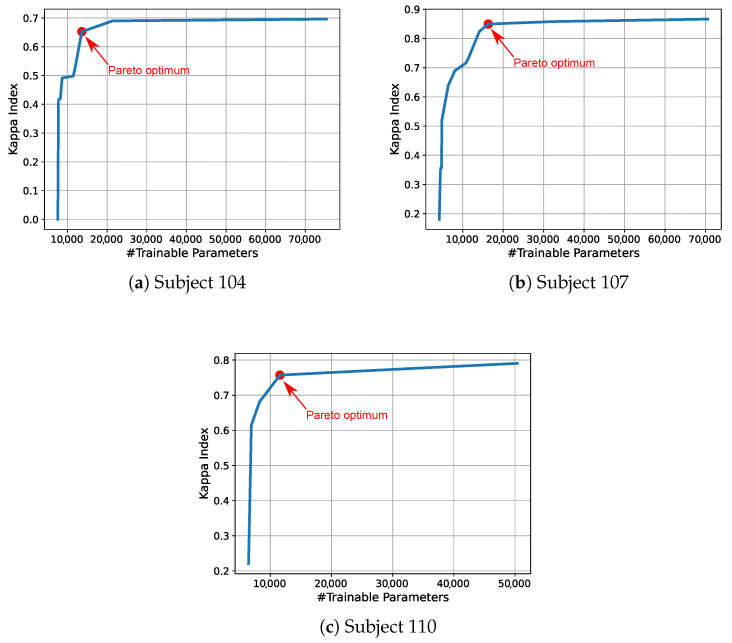
Pareto front corresponding to the neural network optimization process for different subjects.

**Figure 8 sensors-21-02096-f008:**
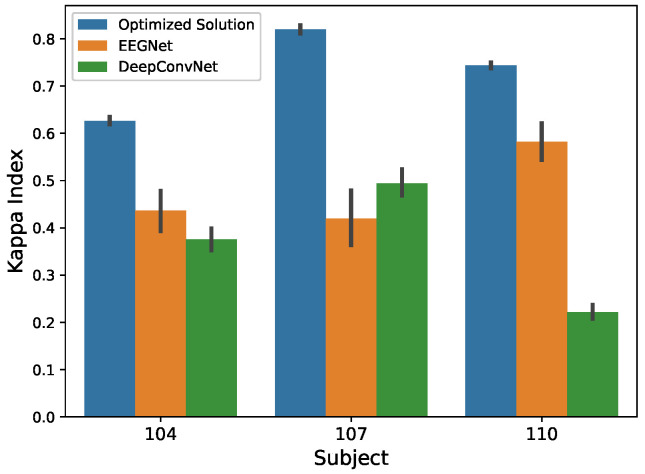
Comparison of the performance that was obtained by different EEGNet models, including the optimized architecture generated by our optimization framework, corresponding to the Pareto optimum.

**Figure 9 sensors-21-02096-f009:**
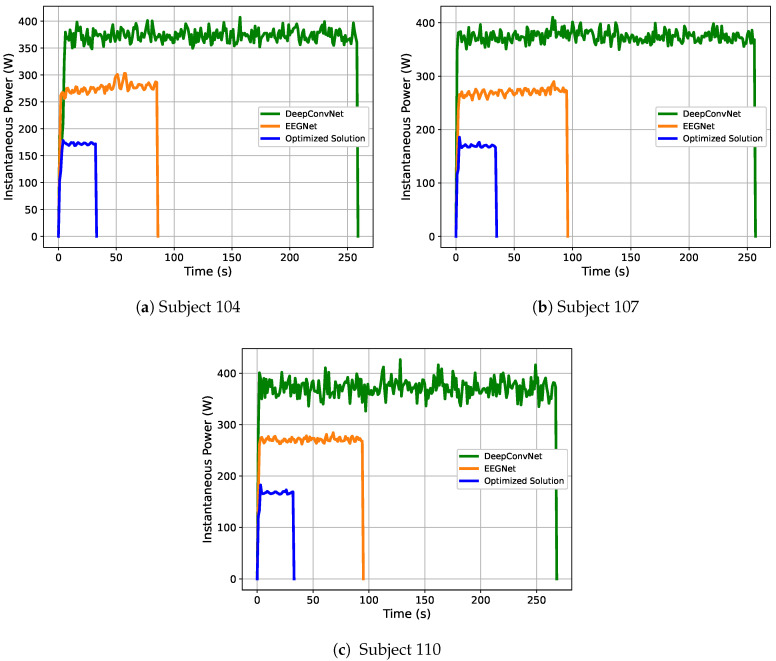
Instantaneous power obtained while training different models and subjects.

**Table 1 sensors-21-02096-t001:** Number of examples for the different movements of each subject.

Subject	Left Hand		Right Hand		Feet		Total	
	Train	Test	Train	Test	Train	Test	Train	Test
104	60	56	66	65	52	58	178	179
107	56	58	57	58	65	63	178	179
110	58	59	60	60	60	60	178	179

**Table 2 sensors-21-02096-t002:** NSGA-II parameters.

Parameter	Value
Chromosome length (genes)	40
Population size	100
Number of generations	100
Mutation probability	0.05
Crossover probability	0.5
Maximum CNN training epochs	500

**Table 3 sensors-21-02096-t003:** Models corresponding to the trade-off point (solution) in the corresponding Pareto front.

Layers	Parameters	Subject		
		104	107	110
Conv2D	activation	selu	sigmoid	selu
	activity_regularizer	None	None	None
	activity_regularizer_type	None	None	None
	activity_regularizer_value	None	None	None
	filters	4	4	4
	kernel_regularizer	None	Yes	None
	kernel_regularizer_type	None	l1	None
	kernel_regularizer_value	None	0.065	None
	kernel_size	(1, 1)	(1, 2)	(1, 1)
	strides	1	1	3
BatchNormalization	batch_normalization	Yes	Yes	Yes
DepthwiseConv2D	activation	relu	tanh	None
	activity_regularizer	None	None	None
	activity_regularizer_type	None	None	None
	activity_regularizer_value	None	None	None
	depth_multiplier	1	1	1
	depthwise_regularizer	Yes	None	None
	depthwise_regularizer_type	l2	None	None
	depthwise_regularizer_value	0.065	None	None
	kernel_size	1	1	1
	strides	3	4	2
BatchNormalization	batch_normalization	Yes	Yes	Yes
Activation	activation	elu	tanh	relu
AveragePooling2D	pool_size	(1, 4)	(1, 1)	(1, 3)
	strides	3	1	3
Dropout	rate	0.50	0.40	0.40
SeparableConv2D	activation	relu	relu	tanh
	activity_regularizer	Yes	Yes	None
	activity_regularizer_type	l2	l2	None
	activity_regularizer_value	0.040	0.025	None
	depthwise_regularizer	Yes	Yes	Yes
	depthwise_regularizer_type	l2	l2	l2
	depthwise_regularizer_value	0.040	0.070	0.045
	kernel_size	(1, 4)	(1, 16)	(1, 8)
	strides	4	4	2
BatchNormalization	batch_normalization	Yes	Yes	Yes
Activation	activation	selu	relu	relu
AveragePooling2D	pool_size	(1, 4)	(1, 3)	(1, 4)
	strides	2	2	3
Dropout	rate	0.55	0.40	0.25

**Table 4 sensors-21-02096-t004:** Classification results and statistical validation.

Subject	Accuracy		Kappa Index		*p*-Values
	Average	Std. Dev.	Average	Std. Dev.	
**DeepConvNet** [26]					
104	0.58	0.03	0.38	0.04	
107	0.66	0.04	0.49	0.06	
110	0.48	0.02	0.22	0.03	
**EEGNet (baseline) [26]**					
104	0.63	0.05	0.44	0.08	
107	0.63	0.06	0.44	0.08	
110	0.72	0.05	0.58	0.08	
**Optimized Solution**					
104	0.75	0.01	0.63	0.01	$p<1.70\times 10^{-6}$
107	0.88	0.02	0.82	0.02	$p<1.69\times 10^{-6}$
110	0.83	0.01	0.74	0.01	$p<1.64\times 10^{-6}$

**Table 5 sensors-21-02096-t005:** Average Power Consumption during Training.

Model	Subject 104 (W)	Subject 107 (W)	Subject 110 (W)
DeepConvNet [26]	367.8	371.5	368.9
EEGNet (baseline) [26]	268.9	262.9	263.9
Optimized EEGNet	158.3	157.3	156.0

## Data Availability

Not applicable.

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
