# Peer review of "Optimization of Deep Architectures for EEG Signal Classification: An AutoML Approach Using Evolutionary Algorithms"

_sensors, 2021, doi:10.3390/s21062096_

Round 1

Reviewer 1 Report

In this paper, the authors propose a framework used to optimize deep learning models for EEG signal classification. The paper is well written and good to read, didactic.

Major comments:
1. Related works are missing. The contributions of the paper/framework should be highlighted.
2. What are the limitations of the proposed framework? They should be explicitly declared, mainly for further investigation.

Questions (answers can/should be used to improve the paper):
3. Electrode positions are according to the 10-20 coordinate system. Why were only 15 electrodes used for collecting data?
4. Consider the sentence "Samples corresponding to 178 BCI patterns were recorded for training and 179 patterns for testing.". I could not understand this sentence. Were there 178/179 labels?
5. Is the sample size composed of only 3 subjects representative? What procedures were performed for proper data collection? For example, criteria for selecting individuals, information for participants not to drink caffeine before data collection, etc.
6. Is the framework proposed in this paper a conceptual model? I mean, as there is no information on implementation aspects, I am supposing it is a conceptual framework, such as a guideline for developers. Or is it an implementation (e.g., a Python notebook) available to be reused? This point is not clear to me.

Specific comments:
7. lines 72-73: "...that provide the best performance in terms of classification performance..." -> performance...performance
8. All abbreviations should be defined at the first occurrence (e.g., GA, etc)

Author Response

We would like to thank the reviewer for the received comments in order to improve the paper. Reviewer’s comments are written in red and our responses are written in green in this revision note. We also provide a version of the paper including these changes, that have been highlighted for an easy check.

Please check the attached file with the detailed responses.

Reviewer 2 Report

The Authors propose a multi-objective evolutionary optimization of Deep Learning models for EEG-BCI signal classification in Motor Imagery. They use two performance metrics: the Kappa Index which measures the multiclass classification performance, and the number of parameters to simplify the classifiers. Results of experiments performed on the dataset that contains three classes of imaginary movements: right hand, left hand, and feet, show that obtained classifiers are computationally efficient and outperform the baseline approaches.

The paper is interesting and worth publishing.

Remarks:

  • Please add information about the numbers of examples for the movements right hand, left hand, and feet for each person. Is your dataset publicly accessible?
  • Please describe the method in more consistent form: in Section 2.4 n is the number of decision variables, m is for the number of criteria. n, m in Figures 4, 5 have other meanings. Moreover, in Fig. 4 is xn-i instead of xi.
  • Elements in chromosome are of different type. Please describe the crossover method.
  • Please describe the rule used to determine the red points in Fig. 7.

Author Response

(The authors gave the same response as above.)

Reviewer 3 Report

The authors proposed a Deep learning structural and hyperparameters optimization method which proposes different architectures due different layer combinations. The proposed techniques utilized the NSGA-II, an evolutionary algorithm for performing multi-objective optimization process. Metrics such as Pareto Front, Kappa Index and CNN Parameters were used for the performance evaluation. Results show that the proposed approach outperformed the baseline techniques such EEGNET and Compact-CNN. Generally, the manuscript is well written and well organized.

However, the authors need to address the following  concerns;

               In section 2.2 of the paper, lines 137-138, the authors mentioned that the activation functions used in the work are discussed. Although most of the activation functions used are discussed, the authors did not mention sigmoid function and scaled exponential linear unit (SELU) which were both used as indicated in the section 3.2, Table 2. Meanwhile, SELU would have been discussed such that it generalises ELU instead of discussing ELU. Similarly, leaky RELU (LRELU) is discussed but its usage could not be found in based on the Table 2.

               In the mathematical definition of ELU in line 152, what happens at z=0? There is a appears to be a typographical error in the equation.

               In section 2.7, line 288-290, the authors stated that the motivation for using NSGA-II is its ability to achieve a good performance with two objectives. However, SPEA2 technique also has similar property as claimed in the same paper [47] refered by the authors. Is there any special justification for selecting NSGA-II other than the mentioned? If the given point is the only justification, the authors are requested to compare the results obtained using NSGA-II with that of using SPEA2. Also, why did the author ruled out the consideration of NSGA-III despite that there appears to be no claim that NSGA-II outperformed it even when considering two objectives?

               In Comparative studies given in the manuscript lines 382-386, the authors have compared with baselines (EEGNet and Compact-CNN). Howbeit, in the paper [26] and [49] refered by the authors, EEGNet model is also named Compact-CNN, could the authors give a clear cut difference between the architecture of Compact-CNN just as that of EEGNet is given if really different from EEGNet?

Author Response

(The authors gave the same response as above.)

Reviewer 4 Report

In this paper the authors have combined DL with GA to optimise the topology and parameters of the DL-NN. The idea is not new, researchers in the past have used GA to optimised the topology of ANN, other have used GA to train the NN by setting the weight instead of using BP. As know, GA is very computing intensive algorithm, however, DL is an extreme computing resources algorithm. Combing the two together, we end up with an algorithms that takes extensive amount of time to come out with the answer. 

The use of DL with EEG signal will also add to the computation time since EEG signal is sampled at very high rate. In all, the suggested algorithm will need super computer with multi processors to run. I am really interested in the algorithm complexity and execution time. Something the authors have ignored. The results are reported and shows improvements in the accuracy with given specific topology which is selected by GA, but it this improvements can be achieved without the need to use GA, a bit of knowledge and few trails can always reach the same or similar results in setting up the DL topology. 

The GA parameters are no standard, the use of 50% crossover means very lazy search. Normally the crossover rate is 80 to 90%. With a population size of 100, and 100 generations, it will be 10000 DL training sessions. Assuming each session takes an average of an hour on super computer, this is nearly 14 months!

Author Response

(The authors gave the same response as above.)

Reviewer 5 Report

The manuscript presents an application of genetic algorithms for hyperparameter tuning of convolutional neural network models used for Electroencephalography (EEG) signal classification. It is generally well organized and shows adequate results. I believe only minor modifications are necessary for the manuscript.

“However, different activation functions are used beyond the linear activation to avoid issues related to the unbounded nature of the linear function, that could range from -inf to +inf” claim should be simplified. Non-linear functions are necessary for artificial neural network models because the composition of linear functions is a linear function. Meaning that, without the activation function, a deep learning architecture is the same as a matrix rotation + bias term.

I was a bit confused with sections 2.4.2 and 2.5. Section 2.4.2 says that “…1D convolutions are used instead of 2D convolutions. An input channel here corresponds to each EEG electrode since it corresponds to a different signal source” which makes sense. However, Figure 3 in 2.5 shows 2D layers (Conv2D, DepthwiseConv2D, etc.). Table 2 also uses 2D. Thus, it is unclear what are the true data dimensions. Information about the batch input dimensions and the whole dataset could be provided as well. I believe these are important descriptions that should be addressed before consideration of the manuscript for publication.

Another information that could be added into the manuscript is the optimizer used for training.

Figure 4 has a circle on the lower left corner that is not described.

“selu” is used in Table 2, but it was not defined.  

Is there a specific reason for four decimals in table 4?

Second-to-last introduction paragraph

Minor comments:

Line 36: please define Electroencephalography (EEG) for the first occurrence in the main text.

Lines 38 to 39: can you quickly describe what are the “partially invasive procedures” and what ECoG does?

Lines 46-47: “The use of feature selection techniques helps to address this issue that usually results in model overfitting.” Please review. I think you meant “this issue” refers the high-dimensional patterns, but it sounds like you might be talking about the limited number of EEG samples.

Line 118: can you briefly describe “BCI pattern” for the reader?

Figure 2: please add x- and y-axis labels.  

Line 187: all other bullet points start with “It”, but Batch Normalization does not. I suggest keeping the same structure for all bullet points (and I think it sounds better without the “It”)

Equations 7 and 8: please define the variables.

Line 239: please reference the pseudo-code in the main text.

Author Response

(The authors gave the same response as above.)

Round 2

Reviewer 1 Report

The authors answered all of my questions and addressed my concerns.

Reviewer 4 Report

The corrections are acceptable, the paper can proceed to the next stage.